# Technical Note: Pondi – a low-cost logger for long-term monitoring of methane, carbon dioxide, and nitrous oxide in aquatic and terrestrial systems

Martino E. Malerba [A,B], Blake Edwards [C], Lukas Schuster [A], Omosalewa Odebiri [A], Josh Glen [A], Rachel Kelly [A], Paul Phan [A], Alistair Grinham [D], Peter I. Macreadie [A,B]

[A] Centre for Nature Positive Solutions, Department of Biology, School of Science, RMIT University, Melbourne, VIC 3000, Australia
[B] School of Life and Environmental Sciences, Deakin University, Burwood Campus, Burwood, VIC 3125, Australia
[C] Leading Edge Engineering Solutions Pty. Ltd. (LEES), Yackandandah, Vic 3749, Australia
[D] School of Civil Engineering, The University of Queensland, St Lucia Qld 4072, Australia

*Correspondence to*: Martino E. Malerba (martino.malerba@rmit.edu.au)

**Abstract.** Understanding the complex dynamics of greenhouse gases (GHGs) such as carbon dioxide ($CO_2$), methane ($CH_4$), and nitrous oxide ($N_2O$) fluxes between aquatic and terrestrial ecosystems and the atmosphere requires extensive monitoring campaigns to capture spatial and temporal variations adequately. However, conventional commercial GHG analysers limit data collection due to their high costs, limited portability, cumbersome weight, and restricted field autonomy. To overcome these challenges, we developed the *Pondi* – a lightweight (0.8 kg) logger from cost-effective components tailored for long-term (weeks to months) continuous monitoring of $CO_2$, $CH_4$, and $N_2O$ concentrations in terrestrial and aquatic environments. Components for a *Pondi* cost approximately USD 750 (or AUD 1,166) and require six hours of specialised labour. The *Pondi* features solar panels for indefinite runtime, Global Positioning System (GPS) for tracking, an Inertial Measurement Unit (IMU) for motion detection, and an optional microcontroller-powered add-on to support self-venting and additional sensors. The *Pondi* can be attached to either floating chambers to measure aquatic GHG emissions, or terrestrial chambers to quantify respiration (with dark chambers) or net primary productivity (with transparent chambers). The *Pondi* is connected to a cloud-based system for real-time data access and remote configuration. The components for the *Pondi* are readily available in most countries, and basic engineering and IT skills are sufficient to assemble the device. By offering a practical, cost-effective, and reliable solution for GHG monitoring, the *Pondi* contributes to efforts to assess and mitigate anthropogenic GHG emissions.

## Keywords

Methane ebullition, climate technology, IoT environmental sensors, field autonomy, portable gas analysers, data cloud integration, autonomous monitoring systems, environmental impact assessment.

**Data availability:** All data and Gerber files for the printed circuit board will be available upon request.

**Code availability:** Not applicable.

**Author contributions:** M.E.M. and P.I.M. conceptualised the research. M.E.M., B.E., A.G., and P.P. developed the *Pondi*. L.S, O.O., J.G., R.K., and P.P. collected and analysed the data. M.E.M. wrote the first draft. All authors contributed to the final draft.

**Competing interests**: The authors declare no competing interests.

**Short summary**

The *Pondi* is a cost-effective, lightweight logger designed for long-term monitoring of carbon dioxide, methane, and nitrous oxide emissions in both terrestrial and aquatic ecosystems. It addresses key challenges in greenhouse gas monitoring by providing an automated, low-cost, solar-powered solution with cloud connectivity and real-time analytics. Its robust design enables deployment in diverse environmental conditions, supporting large-scale, high-resolution emission assessments.

## 1. Introduction

The continued rise in greenhouse gas (GHG) emissions from human activities is intensifying the impacts of climate change. In 2019, global net anthropogenic emissions reached $59 \pm 6.6$ Gt $CO_2$-eq, which is a 12% increase from 2010 and 54% from 1990 levels (IPCC, 2023). The three dominant GHGs—carbon dioxide ($CO_2$), methane ($CH_4$), and nitrous oxide ($N_2O$)—originate from a range of land- and water-based processes and vary in both atmospheric lifetime and warming potential (EPA, 2023; UN Environment Programme, 2023). Aquatic ecosystems play a critical role in the cycling of all three gases. $CO_2$ is exchanged through aquatic primary production, microbial respiration, and organic matter decomposition (Webb et al., 2019). $CH_4$ is produced in anoxic sediments via methanogenesis and released through diffusion or bubble fluxes (Rosentreter et al., 2021; Saunois et al., 2024). $N_2O$ emissions arise from nitrification and denitrification processes in nutrient-rich waters, including wastewater treatment plants, aquaculture ponds, and agricultural drains (Thakur and Medhi, 2019; Hu et al., 2012). Small inland waters contribute disproportionately to these fluxes due to high rates of biological activity and large surface-area-to-volume ratios (Holgerson and Raymond, 2016). Yet, quantifying GHG emissions from aquatic systems remains challenging, with large uncertainties arising from spatial heterogeneity, episodic fluxes, and limited monitoring at scale (Rosentreter et al., 2021).

While satellite and aerial monitoring offer a broad, top-down perspective of GHG emissions, they often fail to capture the fine-scale variability and mechanistic drivers of emissions at the ground level (Boesch et al., 2021). This limitation is particularly problematic in heterogeneous landscapes, such as agricultural mosaics, where GHG sources are spatially variable and often transient (McGinn, 2006). Reliable *in situ* measurements are essential to overcome this gap, as they provide high-resolution, ground-truth data necessary to calibrate and validate satellite-based and airborne models (Kent et al., 2019; Pigliautile et al., 2020). Ultimately, this synergy between *in situ* and remote sensing approaches enables more accurate monitoring, supports targeted mitigation strategies, and enhances confidence in large-scale emission inventories (Janssens-Maenhout et al., 2020).

The current landscape of commercial GHG analysers for *in situ* monitoring has significant limitations (see comparison in Table 1). While accurate and precise, most commercial $CO_2$, $CH_4$, and $N_2O$ analysers have substantial drawbacks in costs, portability, and energy demands (Rodríguez-García et al., 2023). For instance, products from leading companies in this field–such as G2508 and G2509 Gas Concentration Analyzers by Picarro, Ultraportable Greenhouse Gas Analyzer by Los Gatos Research, and LI-7810 and LI-7815 by Li-COR–are capable of measuring gas concentrations at sub-parts per billion levels. However, these devices are expensive, typically USD >50,000. They are also heavy, weighing up to 20 kilograms, and require a power source (e.g., portable generator) to meet their high energy consumption (20-50 W; Rodríguez-García et al., 2023). Alternatively, GHGs can be quantified by sending samples in pre-evacuated vials to a specialised laboratory, eliminating the need to purchase an analyser (Bonetti et al., 2021; Ollivier et al., 2019). However, this approach comes with higher costs per sample, requires personnel in the field for every measurement, and is unsuitable for long-term deployments (>1 week) because of the risk of gas dissolution and oxidation, leading to underestimation of fluxes (Table 1; Thanh Duc et al., 2020).

Developing loggers for greenhouse gases using cost-effective components is a promising compromise to reduce instrument costs, increase replication, and meet the demand for intensive field campaigns (Table 1). Flux chamber studies, a widely used method in GHG research, involve enclosing a defined area of soil, water, or vegetation to measure gas exchange with the atmosphere. These studies are critical for understanding the spatiotemporal variability of GHG emissions, particularly in ecosystems like freshwater systems, which are significant sources of $CH_4$, $CO_2$, and $N_2O$ (Malerba et al., 2022a; Malerba et al., 2022c). Accurate flux chamber measurements provide insights into key processes driving emissions and inform models used for climate change mitigation and policy development (Janssens-Maenhout et al., 2020).

Today, many sensors can be sourced and combined in automatic, lightweight, long-lasting loggers at much lower costs than high-sensitivity commercial models (Bastviken et al., 2020; Dey, 2018; Maher et al., 2019; Morawska et al., 2018; Rodríguez-García et al., 2023; Curcoll et al., 2022; Dalvai Ragnoli and Singer, 2024; Harmon et al., 2015; Sø et al., 2024). While these sensors may lack the sub-ppm accuracy required for direct atmospheric GHG monitoring, they are well suited for flux chamber studies, where gas concentrations increase by orders of magnitude during incubation. This makes them a remarkably cost-effective alternative for capturing accumulation rates within enclosed spaces. Moreover, these sensors can be installed within loggers with standard features like solar panels for indefinite runtime and Internet of Things (IoT) connectivity to the cloud for real-time data monitoring. However, while many individual components of low-cost, autonomous GHG monitoring are now widely available—such as solar power, cloud connectivity, and off-the-shelf sensors—integrated DIY prototypes that combine all these features into a field-ready, multi-gas logger remain exceptionally rare. Most existing systems are limited to single-gas detection and lack the autonomy and connectivity required for effective field deployment. This gap has created a bottleneck in scaling high-resolution GHG monitoring, especially in regions or applications with limited budgets or technical capacity (Thanh Duc et al., 2020).

This article presents the *Pondi*—a novel open-source IoT device designed to monitor $CO_2$, $CH_4$, and $N_2O$ fluxes (Table 1, Fig. 1). Unlike most existing devices, the *Pondi* is optimised for flux chambers deployed in the field, either mounted on floating chambers to monitor aquatic emissions or terrestrial chambers to monitor emissions from the terrestrial biosphere. This work was motivated by entities like the European Union and the U.S.

Environmental Protection Agency being increasingly interested in low-cost GHG monitoring options to improve

their capabilities for collecting data *in situ* at large scales (Borrego et al., 2015; Watkins, 2013).

Table 1: Comparison of greenhouse gas (GHG) monitoring approaches, including IoT loggers (e.g., *Pondi*), traditional GHG analysers, and manual sampling methods.

| | *IoT logger (e.g., Pondi)* | *Traditional GHG analyser* | *Manual sampling and lab analysis* |
|---|---|---|---|
| *Cost-effectiveness* | Low/intermediate equipment costs (USD <1k per unit), low costs per sample | High equipment costs (USD >50k per unit), low costs per sample | Negligible equipment costs, high costs per sample (approx. USD 20 per sample) |
| *Accuracy* | Sufficient for flux chamber studies | High precision, sub-ppm levels | High precision, but risks of gas dissolution and oxidation |
| *Deployment* | Easy, remote-friendly, solar-powered, self-operating | Logistically challenging, power-hungry, personnel-dependent | Personnel-dependent, unsuitable for long-term deployments |
| *Data Management* | IoT connectivity, cloud-based, real-time monitoring | Varies, often manual data transfer | Manual data transfer after lab processing |

## 2. Materials

### 2.1 Overview

The *Pondi* is our cost-effective solution for continuous GHG monitoring in aquatic and terrestrial environments (Fig. 1; see Fig. S1-S3 for onboard printed circuit board designs, and Table S1 for the list of components). The approximate cost of the components for a *Pondi* is around USD 750 (or AUD 1,166) and requires around six hours of specialised labour to assemble. *Pondi* integrates solar panels to sustain operation indefinitely, with an additional panel available for improved performance in low-light conditions, such as winter in Melbourne (Australia) with 9 hours of sunlight at 2-4 kWh $m^{-2}$ $day^{-1}$ (instead of 15 hours at 5-7 kWh $m^{-2}$ $day^{-1}$ in summer). To ensure seamless data management, *Pondi* maintains connectivity to a cloud-based system, reducing reliance on local storage and enabling immediate data access. In case of connectivity loss, *Pondi* transitions to internal storage, initiating a batch data transfer once connectivity is restored. Additional features include GPS for tracking and an Inertial Measurement Unit (IMU) for detecting motion, orientation, and tilt of the device. Finally, *Pondi*'s modular design can connect to an external unit for additional tasks, such as integrating supplementary sensors (e.g., water turbidity, water temperature) or activating an air pump to reset gas concentrations within the collection chamber to environmental levels.

*Materials and components used*

*GHG sensors*: Off-the-shelf sensors measure gas concentrations within the chamber at a user-configurable frequency to calculate fluxes. These sensors are the Figaro TGS2611-E00 for $CH_4$, Sensirion SCD40 for $CO_2$, and Dynament Platinum P/N2OP/NC/4/P for $N_2O$ (Table 2 and S1). These models were chosen because of their low costs, energy efficiencies, and small sizes. Their detection ranges are 0–10,000 ppm for $CH_4$, 0–40,000 ppm for $CO_2$, and 0–1,000 ppm for $N_2O$. Also, they have already been used for field deployment by others (Berthiaume et al., 2020; Demanega et al., 2021; Eugster et al., 2020; Bastviken et al., 2020; Sieczko et al., 2020; Martinsen et

al., 2018). The Sensirion SCD40 sensor also measures temperature and humidity, which are used to calculate fluxes and compensate gas readings for these environmental factors (see Section "Correcting for temperature and humidity").

Table 2: Summary of the characteristics of the gas sensors used in the *Pondi*. The "Gas" column indicates the target gas measured. "Sensor Model" lists the specific sensor and its detection technology (e.g., metal oxide or non-dispersive infrared). "Range" provides the operational concentration range validated for field use, while "Res" indicates the resolution, or the smallest detectable change in gas concentration. "Accuracy" refers to the measurement uncertainty at a representative concentration, expressed both in absolute and relative terms. "Cross-Sensitivities" describes known sources of interference, such as temperature, humidity, or other gases. "MAPE" refers to the Mean Absolute Percentage Error across a typical measurement range, based on field calibration data. "Notes" explain the strategies implemented to correct or compensate for sensor limitations. Finally, "Ref" provides the source of the information, including manufacturer specifications (with hyperlinks) and peer-reviewed publications. For details on the *Pondi* components, see Table S1.

| Gas | Sensor Model | Range | Res | Accuracy | Cross-Sensitivities | MAPE | Notes | Ref |
|---|---|---|---|---|---|---|---|---|
| $CH_4$ | Figaro TGS2611-E00 (MOx) | 0–10,000 ppm | ~0.1 ppm | ± 1.7 ppm at 28 ppm (ca. 6%) | Humidity and temperature. | 8.93% (3–10,000 ppm) | Temperature correction applied using NTC thermistor. Operating RH usually >50%, minimizing humidity effects. Minimal temperature and humidity effects (Fig. 4C). | Figaro manual, Shah et al. (2023) |
| $CO_2$ | Sensirion SCD40 (NDIR + T/RH sensor) | 0–40,000 ppm | 1 ppm | ± 40 ppm at 5,000 ppm (ca. 5%) | Minimal due to NDIR design. | 19.9% (400–10,000 ppm) | Integrated temperature and RH compensation. Sensor underpredicts above 5,000 ppm (Fig. 3B). | Sensirion manual |
| $N_2O$ | Dynament P/N2OP/NC/4/P (NDIR) | 0–1,000 ppm | ~0.1 ppm | ± 50 ppm at 1,000 ppm (5%) | $CO_2$ (~0.05 ppm $N_2O$ per ppm $CO_2$). | 4.96% (0–1,000 ppm) | $CO_2$ correction factor applied. Sensor robust to temperature and RH variation (Fig. 4A and S7). | Dynament manual |

The *Pondi* supports flexible calibration for each GHG sensor. Users can upload new calibration parameters remotely via the cloud interface, allowing for recalibration without physical access to the device. Following manufacturer guidelines, we performed a one-point calibration for $CH_4$ and $CO_2$ under atmospheric conditions, and a two-point calibration for $N_2O$ using both atmospheric concentrations and a high reference concentration (1,000 ppm). This architecture enables users to correct for sensor drift over time or to regularly apply new calibrations, which is particularly beneficial for long-term autonomous deployments in remote environments.

Typically, sensors deliver data in digital format directly to the desired units. However, the output of the Figaro $CH_4$ sensor is in analog format, requiring additional processing to convert these signals into $CH_4$ concentration

values. Following Figaro's hardware implementation guide, we incorporated a temperature compensation circuit on our sensor Printed Circuit Board (PCB) using a Negative Temperature Coefficient (NTC) thermistor. Through trial and error, we fine-tuned this circuit over several PCB iterations to minimise the impact of temperature fluctuations on $CH_4$ readings. The *Pondi* reports the Figaro sensor's resistance via a voltage divider, and this resistance is subsequently processed in the cloud to determine $CH_4$ concentration levels. By performing this conversion in the cloud, we can continuously refine and update the equation as calibration data changes over time.

Using time series data of gas concentrations measured with *Pondi* sensors, it is possible to estimate the total flux of each gas as:

$$F_g(T, P, RH) = \left( \frac{S_g \cdot C_g(T, P, RH) \cdot V}{A} \cdot Z_d \cdot Z_g \right) \qquad \text{(eq. 1)}$$

Where $F_g(T, P, RH)$ is the total gas flux (mg m$^{-2}$ day$^{-1}$) for the gas $g$ (either $CH_4$, $CO_2$, or $N_2O$); $S_g$ is the rate of change in gas concentration within the chamber over time for each gas (ppm hour$^{-1}$); $V$ is the headspace volume in the chamber (m$^3$); and $A$ is the area of the chamber exposed to the water (m$^2$); $Z_d$ is the conversion factor from hours to days (24 hours day$^{-1}$); $Z_g$ is the conversion factor from g to mg (1000 mg g$^{-1}$); and $C_g(T, P, RH)$ is the conversion factor from ppm to mg m$^{-3}$ for each gas, which is calculated based on temperature ($T$), pressure ($P$), and relative humidity ($RH$), as:

$$C_g(T, P, RH) = \left( \frac{M_g \cdot P_d(T, P, RH)}{R \cdot T} \right) \qquad \text{(eq. 2)}$$

Where $M_g$ is the molecular weight of gas $g$ ($CH_4$ = 16.04 g mol$^{-1}$, $CO_2$ = 44.01 g mol$^{-1}$, $N_2O$ = 44.013 g mol$^{-1}$); $R$ is the ideal gas constant (8.314 J mol$^{-1}$ K$^{-1}$); $T$ is temperature (in Kelvin); $P_d(T, P, RH)$ is partial pressure of dry air (in Pa), which is calculated as:

$$P_d(T, P, RH) = P - e(T, RH) \qquad \text{(eq. 3)}$$

Where $P$ is total atmospheric pressure; and $e(T, RH)$ is vapor pressure of water at temperature $T$ and relative humidity $RH$ (in %), calculated as:

$$e(T, RH) = RH \cdot e_s(T) \qquad \text{(eq. 4)}$$

Where $e_s(T)$ is the saturation vapor pressure of water, calculated using the Magnus-Tetens approximation, as:

$$e_s(T) = 610.78 \cdot exp\left( \frac{17.27 \cdot [T - 273.15]}{T - 35.85} \right) \qquad \text{(eq. 5)}$$

We explored the sensitivity of eq. 1 by systematically altering the values of temperature ($T$), atmospheric pressure ($P$), and relative humidity ($RH$) within the formula, based on typical seasonal variations observed in Victoria (Australia). Specifically, we increased $T$ by 30°C, decreased $P$ by 10 kPa, and increased $RH$ from 33% to 99%, while holding all other variables constant. These changes were used to quantify their effect on the calculated gas flux ($F_g$). Results showed that a 30°C increase in temperature raised $F_g$ by 13%, a 10 kPa drop in pressure increased $F_g$ by 10%, and higher relative humidity reduced $F_g$ by 3% (Fig. S4).

*Floating or terrestrial chambers*: The *Pondi* uses a sealed chamber to accumulate GHGs. Typically, the *Pondi* is
installed on 16-litre plastic chambers, but various chamber designs can be used (for details, see section 2.7
Deployment protocol). The chamber can be outfitted with flotation rings to monitor aquatic emissions or inserted
into the ground for terrestrial flux measurements. In both cases, the chamber must be fully sealed to prevent gas
leakage. For terrestrial applications, the chamber is mounted on a 50 cm metal collar that is driven into the soil to
a depth of 5–10 cm. This collar provides structural stability and ensures a hermetic seal between the chamber and
the soil surface, minimizing diffusion losses during flux measurements. For aquatic applications, the chamber is
supported by a custom-designed flotation frame that maintains vertical alignment and ensures the chamber floats
stably at the water–air interface, minimising tilting and providing stability even during windy conditions.

The central connection between the *Pondi*'s sensor module and the plastic chamber is established using a threaded
plastic screw fitted with an O-ring. This O-ring compresses tightly against the chamber's surface when the screw
is secured, creating an airtight seal. Similarly, the $N_2O$ sensor is connected to the chamber via a second hole, using
another threaded plastic screw with an O-ring to ensure a secure and leak-free connection (see photos in Fig. S5).
The flux calculations based on changes in gas concentrations are configurable to accommodate different chamber
volumes and intake areas.

*Electronics enclosure*: All electronics are incorporated into a waterproof enclosure placed on top of the chamber.
A 32 mm hole at the top of the chamber allows the GHG sensors to sample from the chamber space. A 32 mm
nut is threaded over the protruding sensors inside the chamber to join the electronics to the chamber securely. The
$N_2O$ sensor enters the chamber space separately through a second hole on top of the chamber using the sensor
housing offered by the manufacturer (Dynament).

*Power*: Power is provided onboard by four 18650 Li-Ion battery cells charged through a 2 W solar panel on top
of the electronics enclosure. During summer (typically 15 hours of sunlight at 5-7 kWh $m^{-2}$ $day^{-1}$ and max of
120,000 lux), this solar panel provides enough power for the device to monitor gases for at least four months
every hour, or a week every minute. However, in winter (typically 9 hours of sunlight at 2-4 kWh $m^{-2}$ $day^{-1}$ and
max of 80,000 lux), the solar panel can power the *Pondi* to monitor gases for two weeks at hourly intervals (or a
couple of days every minute). For longer deployment, a second 2 W solar panel mounted onto the side of the
electronics can extend the total photovoltaic input power during long periods of low sunlight and power the *Pondi*
for more than two months at hourly intervals (or two weeks every minute).

*Connectivity*: An onboard Mini Peripheral Component Interconnect Express (mPCIe) slot allows connecting to
many different off-the-shelf modems. However, the most effective way to connect a *Pondi* is through the 4G
Category M1 (CAT-M1) modem, which offers low power and long-range performance and allows data-intensive
tasks such as over-the-air (OTA) firmware updates. In remote locations with no CAT-M1 network, the mPCIe
slot can support data transfer through a satellite network, although this option will incur higher costs from network
providers. Alternatively, in areas with WiFi availability, the *Pondi* can also be configured for wireless local area
network (WLAN) connectivity.

*Onboard Microcontroller Unit (MCU)*: The *Pondi*'s functionality is facilitated by a modern MCU, the ESP32-
S3. While a more basic MCU could also be used, the ESP32 enables helpful modern features such as OTA
firmware updates and many flexible general-purpose input/outputs (GPIOs) for integrating additional sensors and
peripherals.

*Backend data ingestion*: *Pondi* loggers maintain continuous communications with a cloud provider (i.e., Amazon
Web Server; AWS) for uploading telemetry, synchronising device settings, receiving downlink commands, and
for various debugging purposes, such as log uploads (Fig. 2). Data from AWS can be used for a front-end website
where users can manage device settings and visualise the data received in real-time, facilitating efficient data
analysis and interpretation. For example, Leading Edge Engineering Solutions (LEES) has developed a front-end
using data from AWS to visualise and manage *Pondi* at https://dashboard.leadingedgeengineering.com.au (Fig.
2).

*External self-venting attachment*: To accurately measure GHG emissions over long periods, flux chambers must
be periodically reset to ambient conditions to avoid gas saturation. Without venting, gas concentrations inside the
sealed chamber can saturate, leading to an underestimation of emission rates (see section 2.3). The *Pondi* can be
connected to a companion microcontroller to manage the air pump for automatic self-venting. This self-venting
attachment is controlled by a control PCB and includes a small 6–12 V direct current air pump with an airflow
rate of 1.5–2.0 L/min through a 5 mm tube (4700 Adafruit Industries LLC; see Table S1 for components). The
pump can be initiated for a venting cycle at user-defined intervals (e.g., once a week).

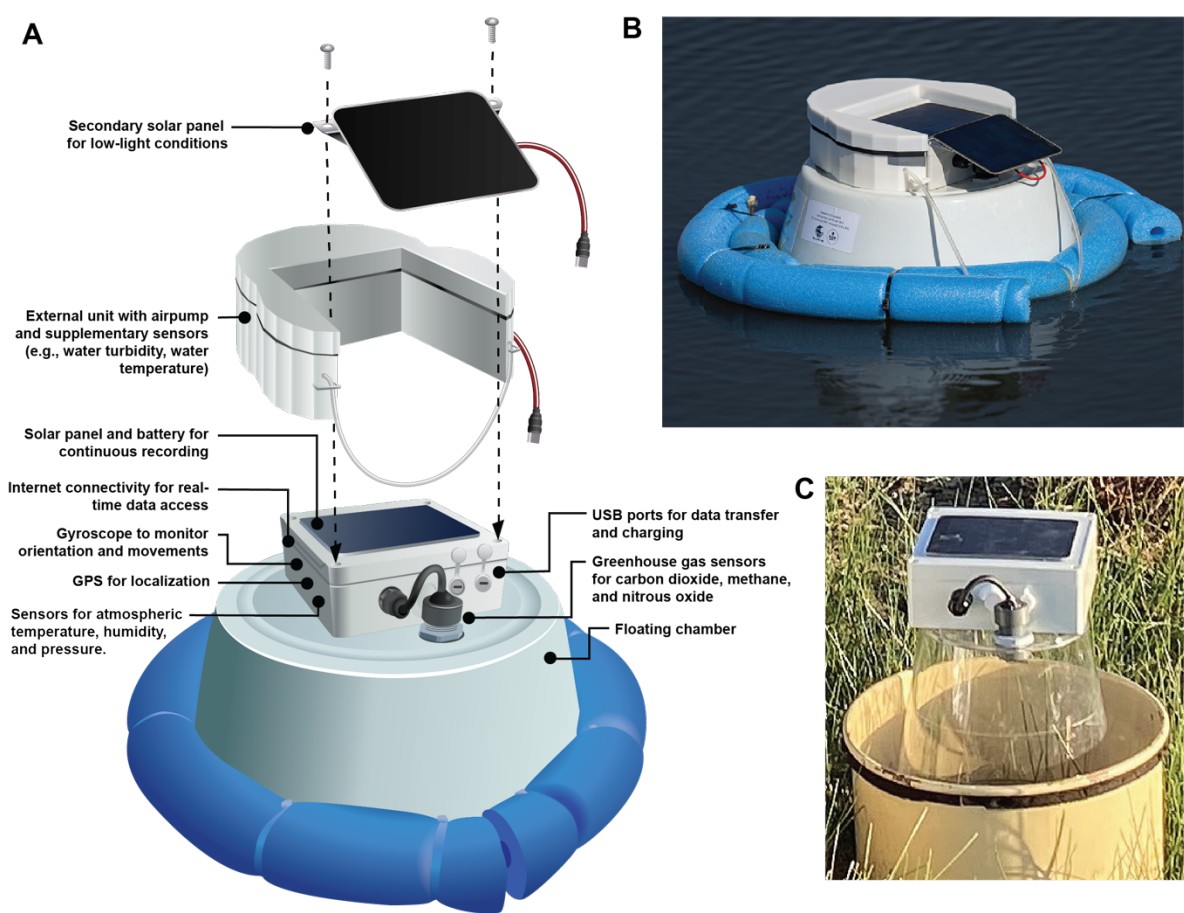

**Figure 1**: *Pondi* logger. (A) Device diagram and labels. Photos of *Pondi* during greenhouse gas monitoring of (B) a water body, including a second solar panel and a secondary unit for periodic self-venting, and (C) a terrestrial system mounted on a transparent chamber hermetically sealed inside a metal collar buried into the ground. See Table 2 for details about the gas sensors, and Table S1 for the list of components. Image credit: (B) Dr Kris Bell, (C) Dr Lukas Schuster.

**2.2 Procedures and data management**

The physical components of the *Pondi* logger, including the microcontroller, sensors, communication modules, and power system, are inside the main enclosure. The microcontroller serves as the central processing unit of the *Pondi* logger, coordinating sensor data collection, data processing, and the operation of other components (Fig. 2). It interfaces with various sensors for $CH_4$ (through an Analog-to-Digital converter; ADC), $CO_2$, $N_2O$, pressure, temperature, and the inertial measurement unit (IMU). Additionally, it can connect to a companion microcontroller to manage the air pump for automatic self-venting and other sensors for measuring water parameters such as temperature and turbidity.

Once the microcontroller processes the data, it transmits this information to the AWS cloud network via a Long-Term Evolution (LTE) modem (Fig. 2). Alternative transmission options include satellite or WiFi connectivity. Upon arrival into the network, functions process alerts, check for erroneous data, and generate system health reports. Users access this data via an online web dashboard where they log in to access data and manage settings. This interface allows users to visualise location and time series data of gas concentrations and fluxes ($CH_4$, $CO_2$, and $N_2O$), along with environmental conditions such as temperature, relative humidity, and atmospheric pressure. It also provides insights into onboard analytics, including battery levels, solar panel charging status, and signal strength. Additionally, the device supports remote configuration via the cloud, enabling users to adjust various settings, such as toggling gas and GPS logging, setting logging intervals, configuring air pump flushing frequencies, sensor calibration, and defining sleep periods to optimise battery usage.

The main enclosure contains the batteries. The device is recharged via solar panel or USB input, with an onboard battery protection system guarding against overcharging, excessive discharge, and other potential risks. The microcontroller monitors the battery state and the flow of charging input to optimise performance by dynamically adjusting the power consumption to suit the conditions.

Additional key features of the onboard logic are:

*Dynamic power usage*: The *Pondi* employs dynamic power management to optimise performance according to available sunlight levels, ensuring efficient energy utilisation. During periods of ample sunlight, power usage is increased to maximise device performance; in low-light conditions, power consumption is minimised to prolong battery life. Key variables governing power usage include the logging rate, upload/reporting frequency, and the duty cycle of sensor heaters. This adaptive approach to power management enables *Pondi* to sustain indefinite operation throughout both summer and winter, facilitated by a single 2 W solar panel in summer and dual 2 W solar panels in winter. The charge sensor monitors the battery's status, optimising power usage and charging cycles, while the charger and battery protection system safeguards against overcharging, discharging, and other potential issues. Finally, the *Pondi* allows the $N_2O$ sensor to be detached when unnecessary, reducing power usage and providing longer battery life.

*Connectivity and onboard storage of offline telemetry*: The *Pondi* is designed to connect with the cloud through the 4G Category M1 (CAT-M1) network to facilitate data offloading immediately after sampling using a commercial data subscription. A SIM card or eSIM with an associated data plan is required for connectivity to this network. This eliminates the need for a separate router or gateway, as the *Pondi*'s onboard modem directly

handles data transmission. In remote locations without CAT-M1 coverage, the device can support alternative
connectivity options, such as WiFi or satellite internet, by connecting additional modules to the mPCIe slot. These
features ensure reliable data transfer in a variety of deployment settings. In instances of network unavailability,
data packets unable to be transferred to the cloud are stored onboard, awaiting reconnection for upload. This logic
can also significantly extend battery life by having an upload rate slower than the logging rate. The modem will
be powered down in between upload events and will upload multiple stored telemetry packets each time the device
is online.

*Movement alerts*: Using the onboard Inertial Measurement Unit (IMU) sensor, *Pondi* can notify users of any
changes in its orientation from handling, lifting, or other movements. These alerts serve as convenient markers to
indicate the commencement and conclusion of a deployment. Furthermore, they offer insights into external
influences, including wind conditions or wildlife interacting with the device – such as birds, mammals, or
amphibians.

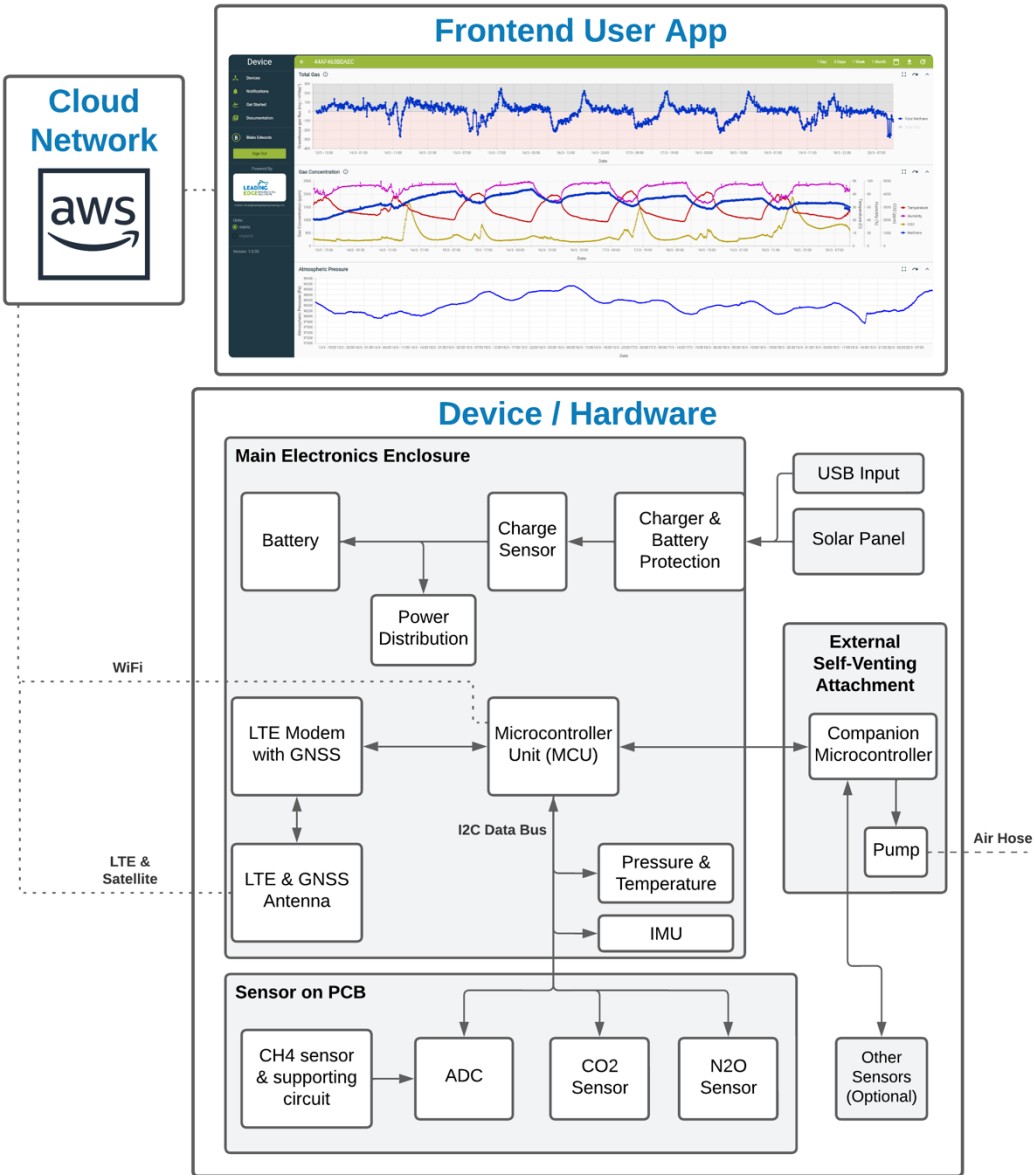

**Figure 2**: Operational logic of the *Pondi* logger. The main enclosure contains the microcontroller, batteries, sensors, and communication modules. The device is powered and recharged through a USB input or a solar panel. It supports connectivity to an external self-venting attachment, which includes a companion microcontroller that controls the automatic air pump and integrates additional sensors. Data from the *Pondi* is transmitted to the cloud via LTE, satellite, or WiFi, enabling real-time monitoring through the frontend user interface.

### 2.3 External self-venting attachment with companion microcontroller

The *Pondi* loggers measure GHG concentrations within a sealed chamber. The concentrations of $CH_4$, $CO_2$, and $N_2O$ inside the chamber rise or fall over time due to release or absorption from soil or water sources. As these gases move between terrestrial or aquatic systems and the air within the sealed chamber, the *Pondi* monitors their concentrations (in ppm) and calculates their flow rates (in mg day$^{-1}$ m$^{-2}$). However, gas accumulation does not continue indefinitely. Eventually, the gas concentrations reach a point where the emission rate equals the diffusion rate. At this equilibrium point, known as saturation, the gas levels stabilise, and the emission rates recorded by the *Pondi* loggers no longer represent the typical emissions of a habitat.

For long-term monitoring of aquatic or terrestrial emissions, the system must be vented periodically (typically once a week) to prevent saturation. Manual venting involves temporarily opening the sealed chamber to equalise the air with atmospheric conditions. For automatic venting, the *Pondi* can connect to an external self-venting attachment, which includes a companion microcontroller that controls an air pump to reset the chamber air to atmospheric concentrations. The microcontroller is programmed to initiate the venting process at user-configurable intervals. During each venting cycle, the air pump operates for a set duration (typically one hour) to flush the chamber with fresh air, ensuring a complete reset to ambient conditions. To prevent pressure buildup inside the chamber, the *Pondi* incorporates a pressure-regulating valve. This valve automatically opens for ten minutes following each flushing event, allowing the chamber to equilibrate with atmospheric pressure. This ensures that the system operates under stable conditions and eliminates potential artifacts in gas flux measurements caused by over-pressurization.

Users can specify the start time, day, and frequency of venting events (e.g., once a week) via the *Pondi*'s cloud-based interface or preset configurations. The algorithm ensures that the venting process aligns with power availability, prioritising periods of sufficient solar energy to recharge the batteries and maintain uninterrupted operation. This automated venting capability enables the *Pondi* to monitor emissions continuously over long-term deployments, minimising the risk of gas saturation while reducing the need for manual intervention.

The companion microcontroller in the external self-venting attachment can also manage additional sensors via a cable extending into the water. These sensors can measure environmental indicators, such as water temperature and turbidity. The companion microcontroller transmits the collected data to *Pondi*'s MCU, preparing it for upload to both the cloud back-end and the user-facing application.

### 2.4 Assessment and sensor validation

We validated the $CH_4$ and $N_2O$ sensors within the concentration range specified by the manufacturer: atmospheric levels to 10,000 ppm for $CH_4$ and 0 to 1,000 ppm for $N_2O$. Because $CO_2$ accumulation in flux studies often exceeds the range of the *Pondi*'s $CO_2$ sensor (400 to 2,000 ppm), we validated its performance outside its specified range (0 to 10,000 ppm). Before validation, we calibrated all sensors using a 2-point calibration for $N_2O$ and a 1-point calibration for $CH_4$ and $CO_2$ (see section 2.1 for details).

We validated the precision and accuracy of all GHG sensors in the laboratory. We created known gas concentrations inside a sealed 15 L plastic water drum (AdVenture Blue Tint Water). For $CO_2$ and $CH_4$, we

introduced pure $CO_2$ and $CH_4$ from commercial cylinders using a high-precision fixed flow regulator at 0.25 L

$min^{-1}$ (PureGas Aust Pty Ltd). We achieved five concentrations from atmospheric level to 10,000 ppm at 2,000

346   ppm increments by opening the regulator at 14-second intervals. We used a commercial greenhouse gas analyser

(UGGA, Los Gatos Research, Model 915– 0011) to check these concentrations. For $N_2O$, we used three gas

cylinders at 0, 500, and 1,000 ppm (all balanced with nitrogen gas) to fill the drum and record *Pondi* readings.

The drum was kept at 21°C and away from sunlight. We exposed the *Pondi* to each concentration for 10 minutes

before taking five measurements every two minutes and using the average value.

For each sensor, we calculated the Mean Absolute Percentage Error (MAPE) as the average magnitude of

percentage errors between predicted ($\hat{y}$) and actual values ($y$), calculated as

$$\frac{1}{n}\sum_{i=1}^{n}\frac{|y_i - \hat{y}_i|}{y_i} \times 100. \qquad \text{(eq. 6)}$$

The $CH_4$ sensor recorded low MAPE (8.93%), demonstrating high precision and low bias overall (Fig. 3A and

Table 2). Only at very high values (ppm > 8,000) did the readings show, on average, 10.6% of systematic

overprediction (see points above the 1:1 line). The $CO_2$ sensor recorded the highest MAPE (19.9%; 3B and Table

2). It performed well within the concentration range specified by the sensor manufacturers and up to 5,000 ppm.

Beyond that, the sensor systematically underpredicted readings by on average 21% (see points below the 1:1 line).

Finally, the model for $N_2O$ had low MAPE (4.96%), partly because this sensor has a smaller range (up to 1,000

359   ppm instead of 10,000 ppm; Fig. 3C and Table 2).

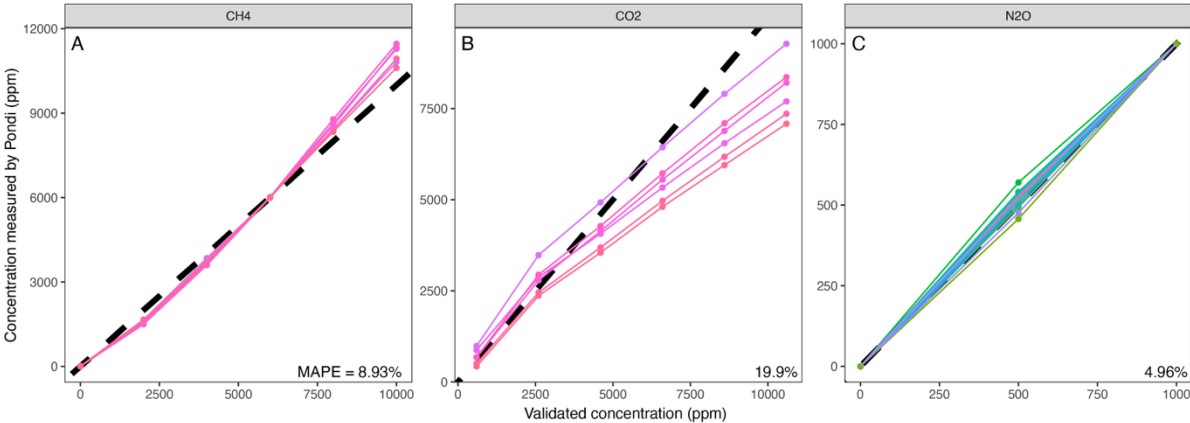

**Figure 3**: Validation of *Pondi* GHG sensors. We tested the $CH_4$, $CO_2$, and $N_2O$ sensors in the *Pondi* at various

gas concentrations using gas cylinders in the laboratory. The dashed line represents the unity line (1:1 ratio). Each

coloured line shows the recordings of a different *Pondi*.

## 2.5 Correcting for temperature and humidity

Previous studies have highlighted potential issues with temperature and humidity affecting sensor signals. To address this, we ensured that all sensors included appropriate corrections for these environmental variables. The $CO_2$ sensor (SCD40) features integrated temperature and humidity sensors, enabling real-time compensation across its operating range. Similarly, the $N_2O$ sensor (Dynament Platinum P/N2OP/NC/4/P) incorporates temperature and humidity compensation as part of its non-dispersive infrared (NDIR) technology. In contrast, the $CH_4$ sensor (Figaro TGS2611) lacks built-in corrections and is known to be sensitive to temperature and humidity (van den Bossche et al., 2017; Bastviken et al., 2020). To mitigate this, we implemented a temperature compensation circuit on the Printed Circuit Board (PCB) using a Negative Temperature Coefficient (NTC) thermistor to minimize temperature effects on $CH_4$ readings. For humidity, while dry conditions (relative humidity <35%) can compromise $CH_4$ sensor reliability (Eugster and Kling, 2012), the $CH_4$ sensor inside the *Pondi* chamber consistently operates at high humidity levels (50–100%), minimizing this concern.

We tested sensor performance for $CO_2$, $CH_4$, and $N_2O$ under controlled laboratory conditions simulating field-relevant temperature and humidity extremes (Figs. 4, S6, S7). Three *Pondi* loggers were placed sequentially in a heated room and a refrigerator to create two scenarios: hot and humid (36°C, 75% RH) and cold and dry (15°C, 50% RH). These conditions reflect the typical range encountered in mid-latitude field deployments. However, future work will include validation under more extreme temperature and humidity regimes, particularly to support applications in tropical and arid environments. Once temperature and humidity reached equilibrium (see shaded regions in Fig. S6), we recorded mean gas concentrations to evaluate sensor accuracy across both extremes.

Results showed that $N_2O$ readings remained consistent across both conditions ($F_{1,4} = 0.139$, $p = 0.73$; Fig. 4A). $CO_2$ readings exhibited a 7% decrease under cold and dry conditions ($F_{1,4} = 7.85$, $p = 0.048$; Fig. 4B). $CH_4$ readings showed no statistically significant differences ($F_{1,4} = 2.08$, $p = 0.22$), though there was a slight 3% decrease in colder and drier conditions (Fig. 4C). These findings demonstrate that temperature and humidity effects on *Pondi* readings are minimal and unlikely to influence estimates in chamber flux studies, where gas concentrations typically increase by several folds. Furthermore, these results align with the precision and accuracy estimates for these sensors (cf. Fig. 3).

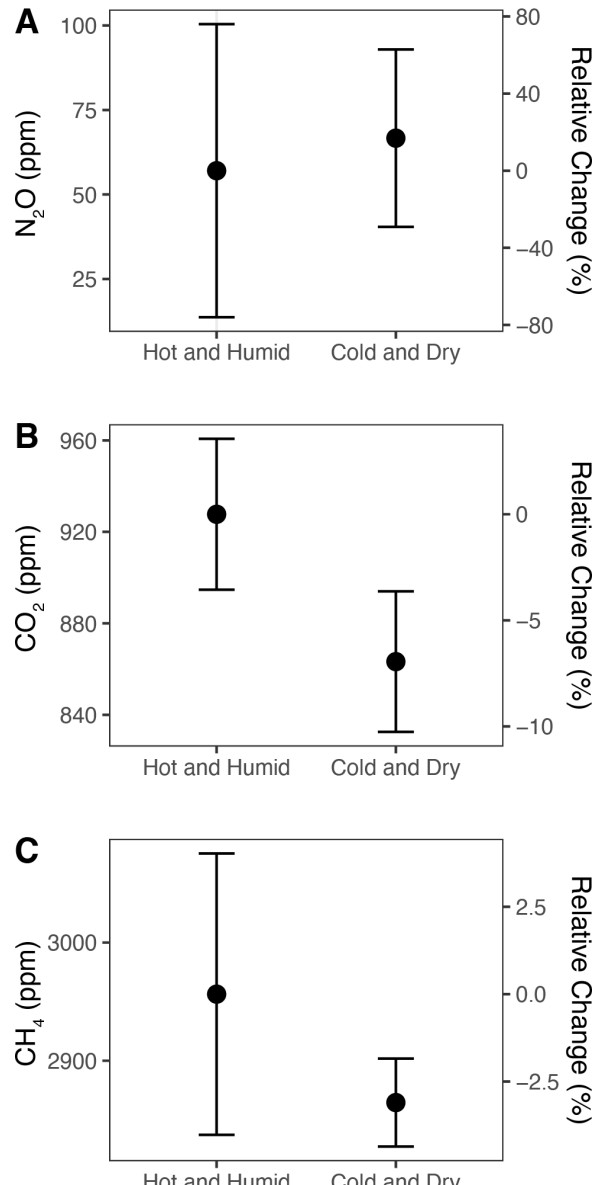

**Figure 4**: Impact of environmental conditions on gas sensor readings, comparing hot and humid conditions (36°C, 75%) with cold and dry conditions (15°C, 50%). Means and confidence intervals are based on data from three *Pondi* units after reaching equilibrium. Refer to Figs. S6 and S7 for detailed time series of all measured parameters during the trial.

## 2.6 Correcting for cross-sensitivities

The manual of the Dynament Platinum $N_2O$ sensor highlights potential cross-sensitivity with $CO_2$, necessitating an approach to account for this effect (Table 2). We used six *Pondi* across two relative humidity levels (medium at 50% and high at 70%) to test how increasing $CO_2$ concentrations might generate false readings for $N_2O$. The results revealed a consistent spurious increase of 0.05 ppm ($\pm$0.002 SE; $F_{1,22} = 706$, $p < 0.001$) in $N_2O$ readings per ppm of $CO_2$, regardless of humidity levels (Fig. 5). To address this, we applied a correction factor based on this relationship to the $N_2O$ data.

The $CO_2$ sensor operates using nondispersive infrared (NDIR) technology, which is intrinsically less susceptible to cross-sensitivities than electrochemical sensors (Table 2). Based on manufacturer specifications and our validation tests, $NO_2$ does not interfere with $CO_2$ detection in this configuration. Additionally, elevated $CO_2$ concentrations had no measurable impact on $CH_4$ readings (Fig. 5).

Outdoor testing demonstrated that the $N_2O$ sensor maintained stable and accurate readings across a broad range of environmental conditions over several weeks. When deployed in a clean plastic bucket filled with rainwater and left outdoors, the *Pondi* consistently reported steady $N_2O$ concentrations, with no detectable influence from fluctuating weather conditions such as temperature, humidity, or solar exposure.

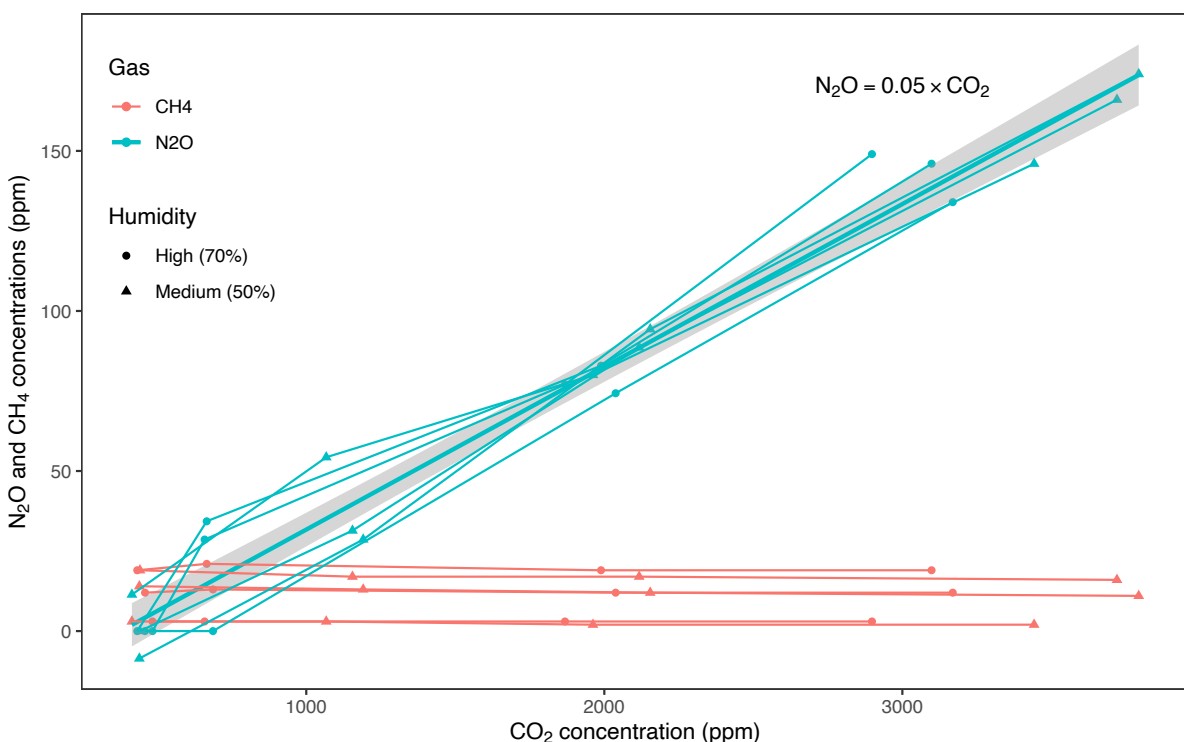

**Figure 5**: Testing the cross-sensitivity of $CH_4$ and $N_2O$ concentrations with increasing $CO_2$ levels. While $CH_4$ was insensitive to $CO_2$, $N_2O$ readings increased by 0.05 ($\pm$0.002 SE) ppm per ppm of $CO_2$, regardless of humidity levels.

## 2.7 Deployment protocol

*Pondi* loggers must be connected to sealed chambers to monitor the accumulation of greenhouse gas fluxes. These flux chamber studies can happen in aquatic or terrestrial systems, and several deployment protocols are possible. Below, we describe our typical setup for aquatic and terrestrial deployments.

*Aquatic Systems*

We installed *Pondi* atop a floating chamber to monitor GHG emissions of aquatic systems (Fig 1B). Before deployment, we activated the logging function through the frontend user app, typically recording data hourly for several weeks. The *Pondi* was carefully placed on the water surface and gently manoeuvred several meters from the shoreline. We anchored the *Pondi* by either tethering with a rope and a 500 g lead sinker, or by installing a pulley system spanning the waterbody, facilitating controlled offshore positioning. In areas with high bird activity such as farm dams, we recommend adding a transparent plastic sheet above the solar panels to shield them from bird droppings (see example in Fig. 6A).

*Terrestrial Systems*

We embedded a 50 cm metal collar 5-10 cm into the soil. An hour later, we affixed a 10-litre plastic transparent chamber inside the metal collar using rubber gaskets to ensure a hermetic seal (Fig 1C). The *Pondi* was on top of the transparent chamber to monitor gas accumulation. To record carbon fluxes while permitting photosynthesis, the transparent chamber was exposed to natural sunlight, with concurrent measurements of temperature and light intensity. For monitoring dark respiration, the chamber was shielded from light using insulation material. Before switching from dark to light measurements, we flushed the gas collection chamber to restart from atmospheric conditions. Dark and light measurements were typically recorded at one-minute intervals for thirty minutes. Recording light intensity and temperature outside the *Pondi* and plant biomass inside the chamber offered valuable data for understanding patterns in dark and light respiration.

## 3. Results and discussion

Here, we present and discuss the results of three case studies in which we used *Pondi* to measure concentrations and fluxes of $CO_2$, $CH_4$, and $N_2O$ in different settings, including agricultural ponds, wastewater lagoons, and freshwater wetland systems.

### 3.1 Case study 1: Agricultural ponds

Small freshwater systems significantly contribute to the uncertainty in global $CH_4$ budgets (Saunois et al., 2024). This uncertainty partly stems from a lack of data at large spatiotemporal scales necessary to capture the main drivers, such as light, temperature, and rainfall (Naslund et al., 2024; Bastviken et al., 2020). Additionally, short-term monitoring of aquatic habitats often underestimates fluxes by neglecting $CH_4$ ebullition—sporadic releases of $CH_4$ bubbles from sediments—which is a major emission source (Grinham et al., 2018).

Agricultural ponds (also known as farm dams, impoundments, dugouts, or excavated tanks) are water bodies used in agriculture for irrigation and livestock (Malerba et al., 2021). They are significant sources of GHGs, emitting more per area than many freshwater systems (Grinham et al., 2018; Ollivier et al., 2018). This emission results from the decomposition of organic matter, influenced by temperature, water level changes, and the presence of nutrients and organic matter (Malerba et al., 2022b). The typical time series of GHG fluxes from farm dams shows rapid increases in $CO_2$, reaching saturation after 2-3 days at around 10,000 to 20,000 ppm (Fig. 6). For $CH_4$, the concentration typically increases linearly for around a week until saturating at around 5,000 to 10,000 ppm (Fig. 6).

The deployment of *Pondi* in farm dams addresses significant logistical and methodological challenges in monitoring GHG fluxes in small agricultural water bodies. These devices can monitor multiple sites for long periods, capturing both ebullitive (sudden release of gas bubbles) and diffusive (gradual release) fluxes to better inform on average emissions and environmental drivers. For example, Odebiri et al. (2024) used *Pondi* to continuously monitor $CH_4$ and $CO_2$ for three months in 20 agricultural ponds in Victoria, Australia. This study analysed seasonal drivers to conclude that fencing farm dams to exclude livestock could reduce $CH_4$ emissions by 72-92%. Moreover, the simple design of *Pondi*, combined with geolocation and cloud connectivity, opens opportunities for citizen science programs. For example, farmers could receive a *Pondi* and only have to put it in the water to start data collection. This approach could further enhance the cost-effectiveness of documenting GHG fluxes at larger scales without relying on field technicians.

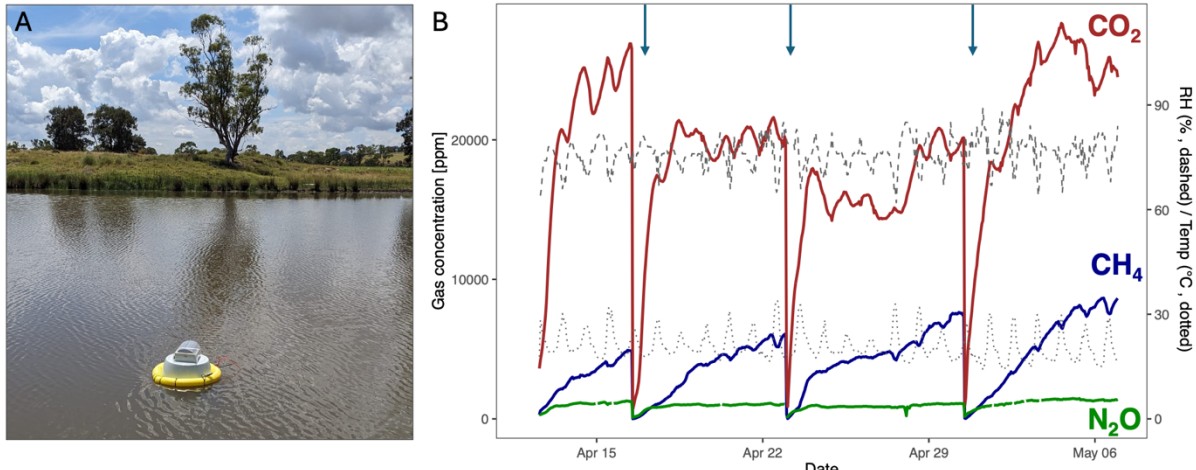

**Figure 6**: (A) *Pondi* in a farm dam. (B) Four weeks of hourly $CO_2$, $CH_4$, $N_2O$, relative humidity (RH), and temperature measurements inside the floating chamber of a *Pondi* in a farm dam. The arrows indicate the three venting events when the air pump diluted gas concentrations by injecting fresh air into the chamber. Image credit: (A) Dr Pawel Waryszak.

## 3.2 Case study 2: Wastewater lagoon

Wastewater treatment plants (WWTPs) emit significant amounts of GHGs, including $CO_2$, $CH_4$, and $N_2O$ (Nguyen et al., 2019). Wastewater typically contains high organic loads and nutrient concentrations, especially nitrogen and phosphorus (Carey and Migliaccio, 2009). These high nutrient concentrations create ideal conditions for microbes to produce $CH_4$ through methanogenesis and $N_2O$ through nitrification and denitrification (Li et al., 2021b). According to IPCC estimates, global GHG emissions from WWTPs account for approximately 2.8% of total anthropogenic emissions (IPCC, 2007). However, these figures are highly uncertain because they were estimated using average emission factors from WWTPs worldwide (IPCC, 2007).

Long-term deployments of *Pondi* loggers in wastewater treatment plants (WWTPs) enable precise quantification of anthropogenic GHG emissions at multiple locations. For example, $CO_2$ and $N_2O$ concentrations monitored by a *Pondi* deployed in a wastewater lagoon rose rapidly, reaching saturation within a day (Fig. 7). In contrast, $CH_4$ accumulated more gradually, saturating after approximately one week. To continue measuring emission patterns, we vented the chamber weekly to reset gas concentrations to ambient atmospheric levels (Fig. 7).

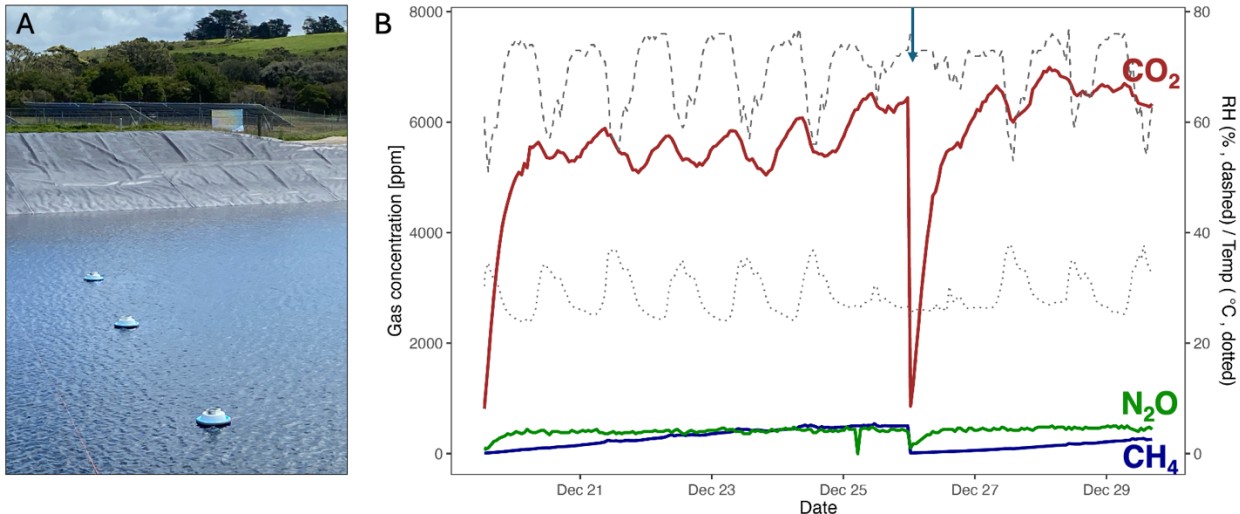

493

**Figure 7**: (A) Three *Pondi* in a wastewater lagoon. (B) Ten days of hourly CO₂, CH₄, N₂O, relative humidity (RH), and temperature measurements inside the floating chamber of a *Pondi* in a wastewater lagoon. The arrow indicates the venting event when the air pump diluted gas concentrations by injecting fresh air into the chamber. Image credit: (A) Dr Lukas Schuster.

### 3.3 Case study 3: Terrestrial fluxes

Measuring GHG fluxes from terrestrial habitats is essential to understanding their role in carbon sequestration, which is influenced by both abiotic and biotic factors (Smith et al., 2014; Rodrigues et al., 2023; Wu et al., 2023). Restoring degraded ecosystems has become a critical nature-based solution to mitigate climate change by enhancing carbon storage and biodiversity (Houghton et al., 2015; Griscom et al., 2017; Schuster et al., 2024).

Flux measurements using *Pondi* can help understand the GHG balances in terrestrial ecosystems and evaluate the effectiveness of ecological restoration. Restoration sites are often remote and difficult to access, making the *Pondi's* small, lightweight design ideal for easy transportation. Additionally, the design of this logger supports various gas collection chambers to measure different types of GHG fluxes. For example, covering the chamber with insulation material or using a dark chamber allows the measurement of CO₂ emissions from ecosystem respiration (dark measurement; Fig. 8A). In contrast, clear chambers can estimate net ecosystem exchange (NEE), which accounts for both CO₂ emissions and uptake through photosynthesis (light measurement; Fig. 8B).

To measure terrestrial fluxes, there are biological constraints when enclosing vegetation in the chamber for extended durations. Specifically, plants show signs of heat stress, especially when sunlight is allowed to penetrate the transparent chamber. The heat buildup inside the sealed chamber can compromise their physiological functions and introduce inaccuracies in gas exchange measurements. To mitigate this, we limited the duration of terrestrial flux measurements to short intervals (typically <30 minutes), ensuring that plant metabolism remained stable (i.e., linear trends in CO₂ concentrations) and avoiding potential artefacts in the data. Active temperature regulation or intermittent venting might extend the measurement duration while minimising heat accumulation and maintaining plant health.

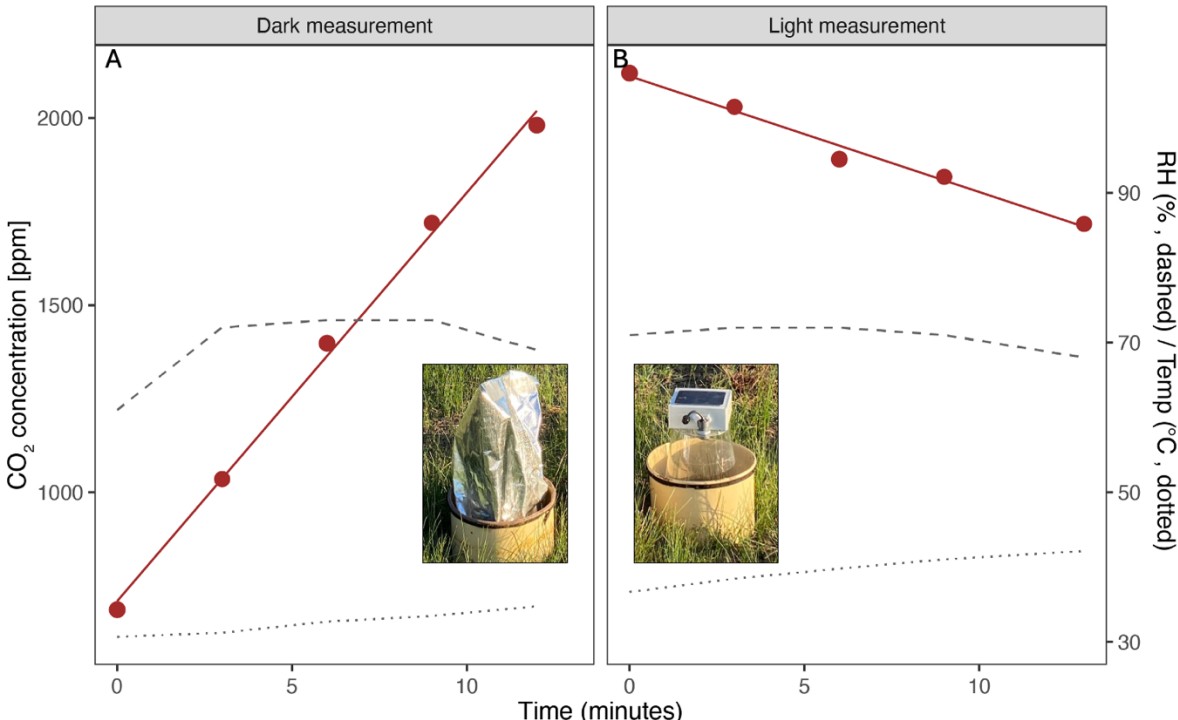

**Figure 8**: Monitoring $CO_2$ concentrations in vegetated terrestrial systems using *Pondi*. (A) *Pondi* recording dark respiration after the transparent chamber is covered with insulation material. (B) *Pondi* recording net primary production by allowing light through a transparent chamber. Coloured dots are measurements from a *Pondi*. Continuous coloured lines are linear models to estimate emission rates (dark measurement) and sequestration (light measurement). Dashed and dotted lines are relative humidity (RH) and temperature measurements inside the chamber of the *Pondi*, respectively. Image credit: Dr Lukas Schuster.

### 3.4 Limitations and further work

Several opportunities exist to enhance *Pondi* loggers for GHG monitoring. First, long-term deployments require regular upkeeping, typically monthly, to clean solar panels and remove biofouling in aquatic systems. Monthly visits provide a natural opportunity to perform routine recalibration, which helps minimise any long-term drift that might otherwise accumulate. However, adding automatic wipers (such as those for underwater cameras and sensors) could reduce maintenance and extend deployment periods. Second, adverse weather may cause the *Pondi* to tip, disrupting GHG capture. Improving chamber design for increased stability could minimise this risk. Third, current sensors in the *Pondi* do not match the accuracy and precision of commercial analysers. Technological advancements could yield low-cost sensors with higher precision and reduced calibration needs. For example, the modern Sensirion SCD40 used in the *Pondi* has significantly advanced $CO_2$ sensor technology, offering higher accuracy and precision at lower costs and smaller sizes than older models (e.g., SenseAir S8, COZIR Ambient $CO_2$ Sensor, Telaire T6615). Fourth, the *Pondi* does not include a fan to mix air in the chamber, as adding one would significantly reduce energy efficiency for long-term deployments. While air mixing has not been an issue in our observations, particularly for aquatic applications, future work could evaluate the benefits of integrating a low-energy fan for terrestrial setups. Finally, the high-frequency sampling capabilities and venting mechanism of

the *Pondi* enable the potential separation of total methane fluxes into their two primary components: diffusive (slow, continuous transport of $CH_4$ across the air–water interface) and ebullitive (episodic release of $CH_4$ bubbles from sediments) fluxes. Although we have not yet conducted this analysis, published methodologies that detect temporal discontinuities in $CH_4$ concentration data are well suited for application to *Pondi* data (Hoffmann et al., 2017; Varadharajan and Hemond, 2012).

## 4. Comments and recommendations

Scalable, low-cost, IoT technology, such as the *Pondi*, can revolutionise our understanding of carbon and nitrogen cycles by reducing costs and overcoming the logistical challenges of collecting data from the field (Salam, 2024; Li et al., 2021a). These large datasets at fine spatial and temporal resolutions will provide the foundation for training complex models. For example, a network of *Pondi* can provide spatially and temporally explicit data to understand complex dynamics in a system. By integrating ground-based measurements with remote sensing technologies, such as drones or satellites, scalable IoT solutions like *Pondi* can unlock transformative insights into ecosystem dynamics, enabling advancements in agricultural productivity, environmental management, and climate resilience at regional and global scales (Shafi et al., 2020; Rajak et al., 2023).

Improving our ability to monitor and predict GHG dynamics can attract private sector investment to advance climate goals (Bellassen et al., 2015). For example, IoT devices can reduce uncertainty and operational costs in carbon projects, offering a robust and transparent system for measurement, reporting, and verification (MRV). This technological approach has the potential to strengthen global carbon credit markets and accelerate climate change mitigation efforts. In addition to carbon monitoring, devices like the *Pondi* can be expanded to include passive acoustic sensors to monitor biodiversity through sound. Using AI-based species recognition algorithms, it is possible to automatically identify birds, frogs, and other vocal fauna, enabling scalable, long-term biodiversity assessments (Pérez-Granados, 2023; Höchst et al., 2022). Integrating AI-driven acoustic biodiversity monitoring with GHG flux data could support the development of joint biodiversity and carbon credit systems, allowing land managers to demonstrate measurable co-benefits of ecological restoration for both climate and nature (Bell and Malerba, 2025).

## 5. Acknowledgements

The Australian Government supported this work through the Australian Research Council awarded to Dr Malerba (project ID DE220100752). The authors thank Drs Pawel Waryszak and Kris Bell for help in the field. We also thank BHP for philanthropic funding for this research. We acknowledge the use of generative artificial intelligence tools to correct grammatical errors and improve the clarity of the text.

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
