# Peer review of "Technical Note: Pondi – a low-cost logger for long-term"

_EGUsphere, 2025_

## Author Comment (AC1)

**We thank Dr Guillem Domènech-Gil for the thoughtful and constructive comments, and we are grateful for the positive feedback on the engineering quality and user-friendliness of the *Pondi*. Below, we respond point by point and outline the changes made to the manuscript accordingly. We will wait to upload the revised manuscript until we get the second reviewer's comments.**

**About calibration and validation**

In section 2.1, you mention the possibility to continuously refine and update the total flux equation as calibration data changes over time while, in section 2.4, you describe one-time calibration (2-point for N2O, 1-point for CH4, and factory pre-calibration for CO2 sensors). This calibration-update feature seems very useful, but if only one-time calibration is needed, which is its purpose?

We agree this point deserved clarification. The "one-time" calibration presented in section 2.4 refers to the initial calibration performed prior to deployment. However, the system architecture allows users to update calibration parameters at any time by uploading new values via the cloud interface. This feature is particularly useful for long-term deployments where sensor drift is expected or if a user performs re-calibration after retrieval. We will clarify this functionality in the revised section 2.1 and add a note in section 2.4 to connect both explanations.

More information on how humidity was controlled during the validation measurements and why the chosen RH and temperature, and CO2, CH4, and N2O concentrations are relevant for the later field measurements would be scientifically valuable and increase future usability of Pondi.

We now provide additional information in the methods (section 2.5). We regulated humidity and temperature during calibration and validation tests by placing three Pondi loggers first into a heated room, and after inside a refrigerator. This approach created two scenarios: hot and humid conditions (36°C, 75%) and cold and dry conditions (15°C, 50%). These conditions cover most of the variability in temperature and humidity typical to mid-latitude field conditions. However, we will also clarify that further validation across more extreme humidity and temperature regimes is planned for future work, especially for tropical or arid applications.

**About interferences, accuracy, and system validation**

The commercial gas sensors used present interferences. While some of these cross-sensitivities are addressed, others remain. In this sense, it would be very interesting for future uses of Pondi to know certain system specifications. The error linked to temperature and humidity, together with the quantified error via MAPE, represents a measurement inaccuracy, different for each sensor and measurement range. Could you clarify the accuracy of Pondi for each sensor and concentration/measurement range? Is it possible to provide MAPE values for the field measurement range?

Great suggestion. In the revised manuscript, we will include a new summary table with:

- Measurement ranges for each sensor
- Resolution
- Accuracy (with references)
- Known cross-sensitivities
- Calculated MAPE values across the most common field measurement ranges (based on our calibration dataset)
- Notes on how we addressed the cross-sensitivities
- Reference for the information provided

The table will look like the one below and we will refer to this in sections 2.4, 2.5, and 2.6

| Gas | Sensor Model | Measurement Range | Resolution | Accuracy | Known Cross-Sensitivities | MAPE | Notes | Ref |
|---|---|---|---|---|---|---|---|---|
| $CH_4$ | Figaro TGS2611-E00 (MOx) | 0–10,000 ppm | ~0.1 ppm | ± 1.7 ppm at 28 ppm (ca. 6%) | Humidity and temperature. | 8.93% (3–10,000 ppm) | Temperature correction applied using NTC thermistor. Operating RH usually >50%, minimizing humidity effects. | Figaro, manual

Shah et al. (2023) |
| $CO_2$ | Sensirion SCD40 (NDIR + T/RH sensor) | 400–40,000 ppm (validated up to 10,000 ppm) | 1 ppm | ± 40 ppm at 5,000 ppm (ca. 5%) | Minimal due to NDIR design. | 19.9% (400–10,000 ppm) | Integrated temperature and RH compensation. Sensor underpredicts above 5,000 ppm. | Sensirion manual |
| $N_2O$ | Dynament P/N2OP/NC/4/P (NDIR) | 0–1,000 ppm | ~0.1 ppm | ± 50 ppm at 1,000 ppm (5%) | Cross-sensitive to $CO_2$ (~0.05 ppm $N_2O$ per ppm $CO_2$). | 4.96% (0–1,000 ppm) | $CO_2$ correction factor applied. Sensor robust to temperature and RH variation. | Dynament manual |

**References**

Figaro TGS2611-E00 manual: https://www.figarosensor.com/product/docs/tgs2611-e00_product%20infomation(fusa)_rev01.pdf

Shah, A., Laurent, O., Lienhardt, L., Broquet, G., Rivera Martinez, R., Allegrini, E., Ciais, P., 2023. Characterising the methane gas and environmental response of the Figaro Taguchi Gas Sensor (TGS) 2611-E00. Atmospheric Measurement Techniques 16, 3391-3419, 10.5194/amt-16-3391-2023.

Sensirion SCD40 manual:
https://sensirion.com/media/documents/E0F04247/631EF271/CD_DS_SCD40_SCD41_Datasheet_D1.pdf?utm_source=chatgpt.com

Dynament P/N2OP/NC/4/P (NDIR) manual: https://www.processsensing.com/docs/dynament/tds0132_1.1Platinum-Dual-Range-Non-Certified-Nitrous-Oxide-Sensor-Data-Sheet.pdf

Section 2.6 provides interesting and important insights, but a relevant question might obfuscate them. Was the CO2 sensor tested against interferences? If CO2 sensor does not present interferences, the CO2 contribution from the NO2 signal can be compensated but otherwise the issue becomes more complex.

We clarify that the $CO_2$ sensor is a nondispersive infrared (NDIR) unit, which is inherently less prone to cross-sensitivities compared to electrochemical sensors. According to manufacturer data and our tests, $NO_2$ does not interfere with $CO_2$ detection in the NDIR configuration used. We have updated section 2.6 to make this clear and provide a citation to the sensor datasheet confirming this.

The field measurements and observations are relevant to validating Pondi, while missing data may induce thoughts of hidden information. To remove this residual possibility, could you include temperature and humidity data in Figures 6, 7, and 8? Do you have long-term terrestrial flux measurements including the different used sensors? Did you notice long-term drift in any of the sensors used? Could saturation values vary over time as seems to happen in Figure 6 and 7?

We have now added temperature and humidity data as additional panels in Figures 6, 7, and 8 to aid interpretation (see below).

While long-term drift is an important consideration, our typical deployment periods thus far have been around one month (see Fig. 6). Although this period is too short to assess long-term drift conclusively, we observed no significant signal decay or instability in any of the sensors during this timeframe. Also, we recommend monthly maintenance visits to clean the sensors and chamber surfaces due to algal buildup and biofouling, especially in aquatic settings. These visits provide a natural opportunity to perform routine recalibration, which helps minimise any long-term drift that might otherwise accumulate.

Regarding long-term terrestrial flux data, we encountered biological constraints when enclosing vegetation in the chamber for extended durations. Specifically, plants showed signs of heat stress, especially when sunlight was allowed to penetrate the transparent chamber to facilitate light-dependent photosynthesis. The heat buildup inside the sealed chamber appeared to compromise their physiological functions and introduce inaccuracies in gas exchange measurements. To mitigate this, we limited the duration of terrestrial flux measurements to short intervals (typically <30 minutes), ensuring that plant metabolism remained stable (i.e., linear trends in $CO_2$ concentrations) and avoiding potential artefacts in the data. Active temperature regulation or intermittent venting might extend the measurement duration in future studies while minimising heat accumulation and maintaining plant health.

[Figure]

**Figure 6**: (A) *Pondi* in a farm dam. (B) Four weeks of hourly $CO_2$, $CH_4$, $N_2O$, relative humidity (RH), and temperature measurements inside the floating chamber of a *Pondi* in a farm dam. The arrows indicate the three venting events when the air pump diluted gas concentrations by injecting fresh air into the chamber.

[Figure]

**Figure 7**: (A) Three *Pondi* in a wastewater lagoon. (B) Ten days of hourly $CO_2$, $CH_4$, $N_2O$, relative humidity (RH), and temperature measurements inside the floating chamber of a *Pondi* in a wastewater lagoon. The arrow indicates the venting event when the air pump diluted gas concentrations by injecting fresh air into the chamber.

[Figure]

**Figure 8**: Monitoring $CO_2$ concentrations in vegetated terrestrial systems using *Pondi*. (A) *Pondi* recording dark respiration after the transparent chamber is covered with insulation material. (B) *Pondi* recording net primary production by allowing light through a transparent chamber. Coloured dots are measurements from a *Pondi*. Continuous coloured lines are linear models to estimate emission rates (dark measurement; red) and sequestration (light measurement; green). Dashed and dotted lines are relative humidity (RH) and temperature measurements inside the chamber of the *Pondi*, respectively.

---

## Author Comment (AC2)

**We thank A/Prof Gerard Rocher-Ros for the thoughtful and constructive comments, and we are grateful for the insightful feedback provided. Below, we respond to each point raised by the reviewer (quotes in grey) and describe how we will revise the manuscript accordingly.**

**As for our understanding of the EGUsphere guidelines, we will wait to upload the revised manuscript until we are authorised by the editor (Prof. Jack Middelburg).**

The article by Malerba and others present a new chamber to measure GHG fluxes from ecosystems, in a very refined design, with telemetry, being one of the most advanced chambers available. The authors also test and present key details on the performance of the equipment, even though some details are missing that I list below.

Thank you for the positive comments.

The only bigger issue I have is a bit more details on the items and estimated cost of a Pondi. Other studies presenting similar chambers (e.g. So et al, 2024; reference below), do a great job with a table of all the main items, sources and rough cost. This would allow a better comparison to other chambers available as well as commercial options, given that the study highlights "low cost" in several places. The same paper provides a repository with more detailed documentation, which could also be necessary to do for this technical note.

Great point. We have now added a new table (Table S1) listing all hardware components and sub-components of the Pondi. We also specified that the approximate cost of the components for a Pondi is USD 750 (or AUD 1,166) and requires around six hours of specialised labour to assemble. This allows for a transparent comparison with other chambers, including the work by So et al. (2024), which we now cite.

Table S1: List of the primary components used in the construction of the Pondi. It includes both core and optional parts. Component: Major subsystem or category of parts (e.g., Enclosure, Solar, Sensors). Description: A brief explanation of the role of each component within the system. Sub-Component: Specific item within the component group. Units per Device: Number of units of that item required for the construction of one Pondi unit. Manufacturer: The company or brand providing the component. Generic items indicate cases where the brand is unimportant. Custom-designed parts (e.g., 3D-printed sensor mounts) were produced by Leading Edge Engineering Solutions (LEES). Items marked as optional (e.g., $N_2O$ sensor, external solar panel) can be omitted to reduce cost or power demand, depending on deployment context.

| Component | Description | Sub-Component | Units per Device | Manufacturer |
|---|---|---|---|---|
| **Enclosure & Mounting** | Protects the internal electronics and | Enclosure | 1.0 | Hammond Manufacturing, 1555RGY |

| | | Vent | 1.0 | Amphenol LTW, VENT-PS1YGY-O8001 |
|---|---|---|---|---|
| | sensors from environmental exposure. Provides a secure housing and mechanical structure for field deployment, including mounting points for floating or terrestrial use. | Chamber | 1.0 | Ezy Storage, 16L Round tbasin |
| | | Pool Noodle | 1.0 | Generic item |
| | | Zip ties | 7.0 | Generic item |
| | | Label - waterproof sticker | 1.0 | Generic item |
| | | Foam seal - Enclosure to PCB (internal) | 1.0 | LEES custom design |
| | | Foam seal - Enclosure to chamber (External) | 1.0 | LEES custom design |
| | | USB-C panel mount waterproof socket & cap | 1.0 | Waterproof IP68 Type C Female to Male PFC Flat Cable 10cm |
| **Solar** | Onboard solar module that recharges the system's battery, enabling long-term autonomous operation without the need for external power sources. | Panel | 1.0 | First Solar, 5V 150mA |
| | | Panel adhesive sealant | 1.0 | Generic item |
| | | Micro-Fit 2 Pin Plug | 1.0 | Molex, 0436450200 |
| **Solar - External (optional)** | An optional, larger solar panel for use in shaded environments or when higher energy capacity is needed (e.g., powering active ventilation or telemetry in low-light areas). | External Panel | 1.0 | Voltaic Systems P126 |
| | | External Panel - USB C plug | 1.0 | LEES custom design |
| | | External Panel - Bracket, 1mm aluminium | 1.0 | LEES custom design |
| | | External Panel - Double-sided tape | 1.0 | LEES custom design |
| | | External Panel - 6mm heat shrink double wall | 1.0 | LEES custom design |
| **PCBs & Components** | Core electronics, including custom-assembled circuit boards, | PCB - Main | 1.0 | LEES custom design |
| | | PCB - Breakout | 1.0 | LEES custom design |

| | | | | |
|---|---|---|---|---|
| | microcontrollers, data storage, and power management systems that run Pondi's operations, read sensors, and handle logging or telemetry. | PCB - Antenna | 1.0 | LEES custom design |
| | | u.Fl cable | 2.0 | TE Connectivity AMP Connectors, 2410329-2 |
| | | Battery holders 18650 | 2.0 | Generic item |
| | | Battery cells | 4.0 | INR18650B |
| | | BG96 mPCI-e | 1.0 | Quectel, BG96 |
| | | mPCie Standoffs | 2.0 | Wurth Elektronik, 9774015151R |
| | | SIM card (cost of each card before data charges) | 1.0 | Generic item |
| | | Micro-Fit 2 Pin Socket | 1.0 | Generic item |
| | | 6-pin sensor cable to breakout PCB | 1.0 | INR18650B |
| **Other Sensors** | Sensors to measure $CO_2$, $CH_4$, temperature, and humidity, critical for calculating gas fluxes. | Methane (CH4) | 1.0 | Figaro TGS2611-E00 |
| | | Carbon Dioxide (CO2) | 1.0 | Sensirion AG, SCD40-D-R2 |
| **Fastners** | Includes bolts, nuts, and screws required to assemble the chamber, secure electronics, and mount components within the enclosure. | M2.5x4 (mPCIe) | 2.0 | Generic item |
| | | M3x6 | 4.0 | Generic item |
| | | M3x12 | 2.0 | Generic item |
| **Printed Parts** | 3D-printed or custom-fabricated parts used to hold sensors, guide airflow, or support other mechanical and structural elements of the system. | Stem | 1.0 | LEES custom design |
| | | nut | 1.0 | LEES custom design |
| | | Battery holders | 2.0 | LEES custom design |
| | | Antenna mount | 1.0 | LEES custom design |
| **Other Consumables** | Miscellaneous materials needed for assembly and | Micro-Fit Pins | | Generic item |
| | | Filament - ABS (kg) | | Generic item |

| Category | Description | Item | Quantity | | Source |
|---|---|---|---|---|---|
| | maintenance, such as adhesives, sealants, tubing, or cable ties, that ensure secure, leak-proof operation. | Conformal coating | | | Generic item |
| **N2O (optional)** | Optional N2O sensor and associated components for measuring nitrous oxide fluxes. May be excluded to reduce cost or power demand if only $CH_4$ and $CO_2$ are of interest. | N2O Sensor | 1.0 | | Dynament Platinum P/N2OP/NC/4/P |
| | | N2O - PCB | 1.0 | | Dynament |
| | | N2O - Panel mount | 1.0 | | Dynament |
| | | N2O - Cable | 1.0 | | 4-core flexible cable |
| | | N2O - 4pin molex plug | 1.0 | | Molex, 0430250400 |
| | | N2O - Gland | 1.0 | | 12mm cable gland |
| | | N2O - Silicon mix | 1.0 | | MG Chemicals Black Flexible Epoxy |
| | | N2O - Petrolium jelly | | | Generic item |
| | | Printed mold | 2.0 | | LEES custom design |
| **Active Venting (optional)** | An add-on module that includes a small pump and microcontroller for periodically flushing the chamber with ambient air to reset internal gas concentrations between measurements. | Pump | 1.0 | | Adafruit Industries LLC, 4700 |
| | | Solenoid | 1.0 | | DFRobot, DFR0866 |
| | | Control PCB | 1.0 | | LEES custom design |
| | | Printed frame | 1.0 | | LEES custom design |
| | | Tubing | 1.0 | | Generic item |
| | | Gland | 1.0 | | 12mm cable gland |
| | | Vent | 1.0 | | 12mm mesh vent |
| | | Vent O-ring | 1.0 | | Generic item |

Finally, the paper mentioned above, are able to separate diffusive from ebullitive fluxes of methane, which is something this study could also explore. Maybe no need to do a new analysis, but mentioning this capacity could be relevant.

Thank you for highlighting this. While the Pondi was designed to capture total net gas fluxes (ebullition + diffusion), the high-frequency sampling capabilities and venting mechanism offer the potential to distinguish between diffusive and ebullitive events based on temporal

discontinuities in CH$_4$ concentration data. Although we have not yet conducted a systematic analysis to separate these flux types, there are published methodologies to do so. We added a statement in the discussion to highlight this potential and suggest it as a priority for future methodological development.

L50: The first paragraph on the different gasses is too broad on the global sources, it could already be narrowed down to the key ecosystems that this study targets. Particularly relevant would be for methane, as half of global emissions are from aquatic ecosystems (Global Methane Budget, Saunois et al 2025, ESSD). The "UN Environment programme, 2023" has a type and is maybe not the best reference.

We agree this section could be more targeted. We revised the paragraph to narrow the scope toward the types of systems targeted by Pondi—namely, small artificial and semi-natural aquatic systems such as farm dams, reservoirs, wastewater lagoons, and vegetated soils. We now cite Saunois et al. (2025, Earth System Science Data) from the Global Methane Budget to better contextualise the importance of aquatic ecosystems in global CH$_4$ emissions. We also replaced the UN Environment Programme citation with Saunois et al. (2025, Earth System Science Data) and Shukla et al. (2022, IPCC).

L74: If spelling out companies like this, would it be needed to provide references for them?

We added info on the specific products by these companies:
- Picarro (e.g., G2508 and G2509 Gas Concentration Analyzer)
- Los Gatos Research (e.g., Ultraportable Greenhouse Gas Analyzer)
- Li-COR (e.g., LI-7810 and LI-7815)

L93: A key reference missing here would be So et al, 2024

We added So et al., (2024, Biogeosciences) in the text.

L198: Some details on the external unit with the air pump are missing: What is the specific design here, which type of fan?

We expanded the relevant methods section to include a description of the external venting unit. It consists of a weatherproof housing that encloses a 5V miniature air pump (4700 Adafruit Industries), controlled by a control PCB to activate the pump for one hour at user-defined intervals (e.g., weekly). The air is filtered and injected into the chamber through a 6 mm silicone tube connected to a dedicated port.

L426: Cannot see arrows in the figure, which are mentioned in the caption.

Thank you for spotting this typo. We have revised the figure to add the arrows indicating the venting events (see below).

[Figure]

**Figure 6**: (A) *Pondi* in a farm dam. (B) Four weeks of hourly $CO_2$, $CH_4$, $N_2O$, relative humidity (RH), and temperature measurements inside the floating chamber of a *Pondi* in a farm dam. The arrows indicate the three venting events when the air pump diluted gas concentrations by injecting fresh air into the chamber.

---

## Author Response (AR2)

Dear Prof. Middelburg,

Thank you for your kind message and for the opportunity to revise our manuscript for *Biogeosciences*. We are very pleased to hear that the paper has been accepted pending technical corrections.

We have carefully addressed the two points you raised:

(1) Line 410: The text incorrectly referred to $NO_2$, and this has now been corrected to $N_2O$.

(2) Figures 6 and 7: We have adjusted the colour scheme to improve clarity and accessibility. We selected a colour-blind friendly palette, using a vivid reddish-orange for $N_2O$ (#F05039), a deep royal blue for $CO_2$ (#1F449C), and a soft desaturated blue-grey for $CH_4$ (#A8B6CC). This combination enhances contrast while maintaining clarity across print and digital formats.

We also greatly appreciated your personal remark regarding your experience with acoustic IR systems. Your observation about the magnitude of the correction factor for $N_2O$ in low-emitting systems is particularly insightful. We have now added a new paragraph at the end of Section 2.6 ("Correcting for cross-sensitivities") acknowledging that the correction introduces substantial uncertainty in $N_2O$ measurements, especially in low-emission environments. We also highlighted this limitation in the conclusions (section 3.4 "Limitations and futher work"), stating that "current sensors in the Pondi do not match the accuracy and precision of commercial analysers, especially for $N_2O$ measurements in low-emitting systems".

Finally, we addressed the editorial comments by (1) using numbers for the affiliation, (2) removing the yellow text in the references, and (3) ensuring there is a manadatory section on "author contribution" (see page 2).

Thank you again for your constructive input and support throughout the review process. We look forward to seeing the paper in print and hope it contributes meaningfully to the field.

Best regards

Martino E. Malerba (on behalf of all co-authors)